# Formulation and Characterization of Mucoadhesive Polymeric Films Containing Extracts of Taraxaci Folium and Matricariae Flos

**DOI:** 10.3390/molecules28104002

**Published:** 2023-05-10

**Authors:** Oana Mihaela Neagu, Timea Ghitea, Eleonora Marian, Laurian Vlase, Ana-Maria Vlase, Gabriela Ciavoi, Pálma Fehér, Annamária Pallag, Ildikó Bácskay, Dániel Nemes, Laura Grațiela Vicaș, Alin Teușdea, Tünde Jurca

**Affiliations:** 1Doctoral School of Biomedical Sciences, University of Oradea, 1 Universității Street, 410073 Oradea, Romania; oana_mihaela13@yahoo.com (O.M.N.);; 2Department of Pharmacy, Faculty of Medicine and Pharmacy, University of Oradea, 1st December Square 10, 410028 Oradea, Romania; 3Department of Pharmaceutical Technology and Biopharmaceutics, University of Medicine and Pharmacy, 8 Victor Babeș Street, 400347 Cluj-Napoca, Romania; 4Department of Pharmaceutical Botany, Iuliu Hațieganu University of Medicine and Pharmacy, 8 Victor Babeș Street, 400347 Cluj-Napoca, Romania; 5Dental Medicine Department, Faculty of Medicine and Pharmacy, University of Oradea, 1st December Square 10, 410073 Oradea, Romania; 6Department of Pharmaceutical Technology, Faculty of Pharmacy, University of Debrecen, Nagyerdei Körút 98, H-4032 Debrecen, Hungary; 7Faculty of Environmental Protection, University of Oradea, No. 26 General Magheru Avenue, 410087 Oradea, Romania; ateusdea@yahoo.co.uk

**Keywords:** Taraxaci folium, matricariae flos, antioxidants, anti-inflammatory, gingivitis, bioadhesive film

## Abstract

Taraxaci folium and Matricariae flos plant extracts contain a wide range of bioactive compounds with antioxidant and anti-inflammatory effects. The aim of the study was to evaluate the phytochemical and antioxidant profile of the two plant extracts to obtain a mucoadhesive polymeric film with beneficial properties in acute gingivitis. The chemical composition of the two plant extracts was determined by high-performance liquid chromatography coupled with mass spectrometry. To establish a favourable ratio in the combination of the two extracts, the antioxidant capacity was determined by the method of reduction of copper ions Cu^2+^ from neocuprein and by reduction of the compound 1.1-diphenyl-2-2picril-hydrazyl. Following preliminary analysis, we selected the plant mixture Taraxaci folium/matricariae flos in the ratio of 1:2 (*m*/*m*), having an antioxidant capacity of 83.92% ± 0.02 reduction of free nitrogen radical of 1.1-diphenyl-2-2picril-hydrazyl reagent. Subsequently, bioadhesive films of 0.2 mm thickness were obtained using various concentrations of polymer and plant extract. The mucoadhesive films obtained were homogeneous and flexible, with pH ranging from 6.634 to 7.016 and active ingredient release capacity ranging from 85.94–89.52%. Based on in vitro analysis, the film containing 5% polymer and 10% plant extract was selected for in vivo study. The study involved 50 patients undergoing professional oral hygiene followed by a 7-day treatment with the chosen mucoadhesive polymeric film. The study showed that the film used helped accelerate the healing of acute gingivitis after treatment, with anti-inflammatory and protective action.

## 1. Introduction

Medicinal plants are an important source of bioactive compounds and antioxidant substances for the human body [1]. Topical herbal therapy is now successfully used [2]. According to the World Health Organization, more than 80% of the population prefers herbal remedies, so medicinal plants are the source of many of today’s medicines [3].

*Matricaria chamomila* L., also called German chamomile, is a plant often found in pharmaceutical preparations and has many beneficial health effects. Extracts from the flowers of this species are used both orally and topically in the treatment of pain, bacterial infections, and mouth sores, but also for respiratory and digestive disorders [1].

Chamomile is known for its anti-inflammatory, analgesic, and antimicrobial properties due to the flavonoids, polyphenols and essential oils in its composition [4].

Dandelion, *Taraxacum officinale* L., is a widespread species that contains numerous bioactive compounds, such as flavonoids and polyphenols, with known diuretic and anti-inflammatory properties [5].

We have chosen these herbal products because they are endemic plants, affordable and known in the literature. The two species in the paper contain considerable amounts of antioxidants, which is why they may be a promising therapy for the prevention and healing of mouth lesions [6,7].

According to the 10th edition of the European Pharmacopoeia, buccal films fall into the class of mucoadhesive preparations. They usually contain hydrophilic polymers, which in contact with saliva turn into a hydrogel that adheres to the mucosa [8].

Mucosal films may be preferred for therapeutic purposes because of the comfort they offer. Their flexibility helps to position them easily in the desired place, unlike other pharmaceutical forms such as mucoadhesive tablets. The adhesion to the buccal mucosa makes these films preferred over oral solutions or oral gels, which are usually easily removed by saliva, and thus have less retention time on the mucosa. The mucoadhesive properties do not allow the film to be easily swallowed or inhaled, the danger of choking is minimised [9].

For the purpose of using mucoadhesive films for mouth wounds, these films help to protect the mucosal lesion, thereby helping to reduce pain and speed the healing of the lesion [10].

Rapid removal of the polymer film due to saliva washout or food ingestion may result in the need for more frequent application. The uneven distribution of the drug over the entire oral mucosal surface may be an inconvenience in the case of extensive lesions. Another disadvantage of using polymer films may be unfavourable patient compliance, which may occur due to unpleasant taste, possible irritation, or foreign body sensation in the mouth [9].

Gingivitis is one of the most common periodontal diseases and can cause severe damage to the oral cavity. Gingivitis manifests through inflammation of the gums, which, if left untreated, can trigger periodontitis in some patients [11].

The aim of this study was to obtain a mucoadhesive polymeric film containing a combination of extracts from chamomile flowers and dandelion leaves for use in patients with acute gingivitis following professional oral hygiene.

In this paper, we have based our work on the hypothesis that an extract containing a mixture of both plants could have a synergistic effect with great benefits for gingival inflammatory processes. The curative effects of extracts from Taraxaci folium and Matricariae flos have been tested separately by various authors with very good results [12,13].

According to expert studies, the polymer used to prepare these films, polyvinyl alcohol, is a safe compound for pharmaceutical formulation, is not mutagenic or clastogenic and does not accumulate in the body [14]. 

Also, mucoadhesive films with phytocompounds are not known in therapy, in the current article the in vivo pharmacological activity of these preparations is evaluated. Due to the advantages presented, the polymeric films studied and presented by us present a novelty in medical practice. 

## 2. Results and Discussion 

### 2.1. Identification and Quantification of Active Ingredients in Plant Extracts

#### 2.1.1. Total Amount of Polyphenols and Flavonoids in Extracts

Flavonoids are a subclass of the polyphenol class and are the main constituents of medicinal plants, with numerous in vitro and in vivo bioactivity studies proving antioxidant and antiproliferative effects [15]. The total flavonoid content (TFC) and total polyphenol content (TPC) were determined in both individual extracts and mixtures obtained. The values obtained were close and were expressed as mg Quercetin (QE)/100 g dry sample (DW), respectively, and mg Gallic acid equivalents (GAE)/100 g dry sample (DW). It can be seen from the obtained results that the extract from dandelion leaves shows higher values in both cases, unlike the extract obtained from chamomile flowers. Statistical analysis (Table 1 and Table 2) to compare the results is provided in Appendix A.

Compared to other studies in which alcoholic extracts from chamomile flowers were analysed we obtained a lower value of polyphenols. TPC in comparative studies were 21.4 ± 0.327 mg GAE/g for an ethanolic extract of chamomile, in which the plant product was purchased from a local shop [16]. For a methanolic extract of Italian chamomile, the TPC value was 2689.2 ± 15 mg GAE/100 g DW [17]. However, values closer to ours were also obtained, such as 3.5 ± 1.7 mg GAE/g DW for an ethanolic extract of chamomile, using an extraction method of solvent stirring followed by evaporation and drying in a desiccator to constant weight [18]. In another study, the methanolic extract of chamomile flowers was subjected to alcohol evaporation to obtain a dry extract, in which an amount of 31.9 mg GAE/kg DW was determined [19].

TFC ranged from 530.9 ± 20 mg QE/100 g DW to 710.7 ± 9 mg QE/100 g DW in a methanolic extract from chamomile flowers, harvested from southern Italy, consisting of much higher values than those obtained by us [17]. The amount of flavonoids in an ethanolic extract of chamomile was 157.9 ± 2.22 mg QE/g dry extract [16]. 

For an extract obtained from dandelion leaves in ethyl acetate by stirring for 24 h, TPC was 10.2 mg GAE/g DW [20], respectively 15.5 mg GAE/g DW for a concentrated methanolic extract, these being values higher than those obtained by us [21]. TFC from the dandelion extract was also evaluated in rutin equivalents at 6.87 mg RE/g DW in a previous study [22]. 

#### 2.1.2. Identification and Quantification of Phytosterols

Several studies attest to the antioxidant activity of phytosterols [23,24]. Among the sterols analysed, ergosterol, stigmasterol, β-sitosterol and campesterol have been identified and quantified in considerable amounts in plant extracts. They are present both in the extract from chamomile flowers—Matricariae flos (M) as well as in the extract from dandelion leaves—Taraxaci folium (T), in the amounts shown in Table 2.

In the comparative studies reviewed, traces of phytosterols were detected in the composition of chamomile without being identified and quantified [25]. In the dandelion extract, stigmasterol, β-sitosterol and campesterol were previously identified [26].

#### 2.1.3. Identification and Quantification of Tocopherols

Vitamin E is an essential vitamin with a strong antioxidant and anti-inflammatory effect and includes several structurally similar compounds, including tocopherols. Following the chromatographic determination, three tocopherols were quantified in Matricariae flos extract and only two in Taraxaci folium extract. Their values are shown in Table 2. 

By the chosen extraction method, we obtained a content of 134.50 ng/mL of α-tocopherol compared to other methods, respectively, by ultrasonication in chamomile extract, a 120.46 µg/mL [27] was found.

Similarly, α-tocopherol (10.64 ± 0.59), γ-tocopherol (10.63 ± 1.66) and δ-tocopherol (3.50 ± 0.00) were detected in fresh leaves of *Taraxacum officinale* L. in a previous study, expressed in µg 100 g^−1^, consisting of higher amounts than those obtained by us [28].

#### 2.1.4. Identification and Quantification of Methoxylated Flavones

Methoxylated flavones belong to the flavonoid class and are characterised by the protection of the hydroxyl fragments by methyl groups. They are lipophilic in character and are thought to act as prodrugs, which after demethylation enhance post-absorption bioavailability. Methoxylated flavones show anti-inflammatory and antiproliferative capacity [15]. The presence of methoxylated flavones detected by HPLC-MS was different in the two extracts. Hispidulin was quantified in both plant extracts, being present in a higher amount in the Matricariae flos extract.

Hispidulin was identified in a previous study in the composition of freeze-dried chamomile extract, estimated at 1.584 ± 0.181 mg/g extract, a value close to that obtained by us. In the same study, values ranging from 0.231–17.060 ± SD mg/g extract were obtained in the fractions of chamomile extract obtained from the gross extract [29].

#### 2.1.5. Identification and Quantification of Polyphenols

Of the polyphenols analysed, some were identified only qualitatively. It can be seen that there are common ones, but also different polyphenols in the extracts. Analysing the results, it can be seen that caffeic acid, chlorogenic acid, p-coumaric acid, ferulic acid, rutozide, luteolin, and apigenin were identified in both extracts, but only chlorogenic acid and luteolin were quantified in both extracts studied.

The following polyphenols were identified in chamomile flower extract: gentisic acid, caffeic acid, chlorogenic acid, p-coumaric acid, ferulic acid, rutozide, quercitrin, quercetol, patuletin, luteolin, kaempherol and apigenin. Of these, only a few were also identified by the UV (ultra-visible) method and quantified, as shown in Table 2.

Following the literature data, the following amounts of polyphenols can be observed in aqueous extracts obtained from chamomile tea: luteolin 0.04–0.13 mg/L, apigenin 2.24–2.60 mg/L, caffeic acid 0.41–1.53 mg/L, rutin 0.26–4.21 mg/L, chlorogenic acid 0.04–4.36 mg/L, p-coumaric acid 0.03–0.11 mg/L [30]. These values are close to those obtained by us in the current study. In another comparative study, the following polyphenols were identified and quantified in the composition of chamomile methanolic extract, in lower amounts than those obtained by us: chlorogenic acid (8.180), caffeic acid (1.296), rutin (6.013), luteolin (5.113), apigenin (1.388) expressed in µg polyphenol/100 µg extract [31]. Apigenin (0.298 ± 0.027 mg/g extract) and luteolin (4.617 ± 0.616 mg/g extract) were determined in freeze-dried chamomile extract with lower values than in the present study [29].

Caftaric acid, caffeic acid, chlorogenic acid, p-coumaric acid, ferulic acid, rutozide, luteolin, and apigenin were identified in dandelion leaf extract. Of these, caftaric acid, chlorogenic acid, p-coumaric acid, rutozide, and luteolin were detected in both ultra-visible (UV) and mass spectrometry (MS) methods, as shown in Table 1. Luteolin is a flavonoid in dandelion extract that inhibits nitric oxide and prostaglandin E production, an effect studied on macrophage cell cultures [32].

Depending on the concentration of ethyl alcohol used to obtain the extract, different polyphenols were determined by the HPLC method in dandelion leaves. Thus, the following polyphenols were quantified in higher quantities than those obtained by us: chlorogenic acid 18–37 mg/100 g DW, caffeic acid 15–22 mg/100 g DW, p-coumaric acid 12–88 mg/100 g DW and ferulic acid 97 mg/100 g DW [33]. Another study quantifies the polyphenols analysed in dandelion leaves as follows: caffeic acid 113.7 ± 12.4 mg%, ferulic acid 7.5 ± 2.1 mg%, p-coumaric acid 3.9 ± 0.6 mg%, being higher values than those obtained by us [34].

Polyphenolic acids are a subcategory of the class of polyphenols, which have a carboxyl group in their chemical structure. They are among the main phenolic compounds found in plants and have important antioxidant and anti-inflammatory activity. The most common polyphenolic acids are syringic acid, vanillic acid, protocatechuic acid, and p-hydroxybenzoic acid [35]. HPLC-MS analysis identified and quantified syringic, protocatechuic and vanillic acid in chamomile extract and only syringic acid and protocatechuic acid in dandelion extract (Table 2).

Compared to other studies on the presence of phenolic acids in dandelion leaves, vanillic acid, or syringic acid were not detected. At the same time, in a previous study catechin was quantified with 5.179 µg/100 µg extract, compared to the current study, in which we did not identify this polyphenolic acid [31].

### 2.2. Antioxidant Activity of Plant Extracts

Cuprac and DPPH methods were used to evaluate antioxidant activity. The antioxidant character was determined for both individual extracts and mixtures of extracts. Close values were obtained for both methods performed. Among the two plant products, it can be seen that Matricariae flos extract has higher antioxidant potency than Taraxaci folium extract, and of the mixtures analysed, the antioxidant character is more intense in the mixture containing Matricariae flos in a higher ratio. The results were expressed in Trolox equivalents. Statistical analysis (Table 3 and Table 4) to compare the results is provided in Appendix A.

The inhibition percentage of other previously studied chamomile extracts ranged from 84.2 ± 0.86 to 94.8 ± 0.03 [16], which are similar levels to those obtained in the present study. By the Cuprac method, the dandelion extract shows greater antioxidant character compared to the current study, obtaining values of 97.1 ± 0.1 mM TE/g DW for the 95% ethanol extract, 180.1 ± 1.5 mM TE/g DW for the aqueous extract, respectively, 407.8 ± 7.5 mM TE/g DW in the 50% ethanol extract [33].

### 2.3. Polymeric Films Reference

#### 2.3.1. Film Display and Sensory Analysis

An ideal mucoadhesive oral film (MOF) should be flexible, elastic, and soft, but at the same time resistant to rupture under the action of mechanical forces. These films must have good bioadhesive strength in order to be retained on the mucosa for the pertinent effect. Swelling of the mucoadhesive film, if it occurs, should not be too extensive, in order to prevent discomfort [36].

Taraxaci folium/Matricariae flos mixture in a 1:2 ratio was used in the preparation of polymeric films with plant extract. The characteristics of the obtained polymer films were determined visually, using the optical microscope and Hedonic test, and can be seen in Table 4 and Figure 1, respectively.

In other studies, in which mucoadhesive films based on PVA 10% and glycerine 5% were studied, they were translucent or opaque in the presence of the drug substance. In this case, the films without the drug were sticky and those with the drug did not show this inconvenience [37]. 

The colour of our films was conferred by the plant extract. In other studies, the colour of the films was assessed using spectrophotometry. The smell of other herbal mucoadhesive films was characteristic of the plant product, with a citrus aroma, and the taste was evaluated as bitter or minty [38]. According to other studies, mucoadhesive films with herbal extracts exhibited green colouration and smooth texture [39].

#### 2.3.2. Film Samples Chromatic and Imagistic Analysis

The initial image that contains all the scanned film samples is presented in Figure 2—the sample codes are inserted for each film samples. Each film sample has two or four film parts organised by columns. Chromatic classes codes and false rendering colour are presented in Table 5. The point cloud of initial image (i.e., Figure 2) and the boxes corresponding to each chromatic class are presented in Figure 3—the CIE L*a*b* trichromatic colour space was used. 

The segmented image is presented in Figure 4. The film sample without extract (M1 and M2) is fully discriminated than the films with extract (C1, C2, C3, and C4)—with one exception, the second part of the C2 film sample which contains a transversal fold that performed a strong specular reflection with colour degradation. As a consequence, the M1_M2 class proportion for this film sample is 23.997%, the highest value among the films with the extract. All other film samples with extract (C1, C2, C3, and C4) performed M1_M2 class proportion under 0.415% compared with the above 99% proportion performed by the film sample without extract (M1 and M2). 

The ExtrCL1, 2, 3, and 4 classes were built up (in this order) with the same a* and b* ranges, but with decreasing L* values. The films with high values for ExtrCL1 class have a lower concentration of extract, and the concentration increases with ExtrCL2 till ExtrCL4 (see Table 6, the false rendering green colour grading). All films with extract present higher proportion values for ExtrCL1, 2, 3, and 4 classes compared with the films without extract. The highest proportions from all C1, C2, C3, and C4 film samples with extract are performed for ExtrCL4, in the range of 74.8% to 89.9%, while the proportions for M1 and M2 film samples without extract range between 0.005% and 0.027% (Table 7). This fact prescribes full discrimination between the C1, C2, C3, and C4 samples and M1 and M2 samples. The same behaviour is performed for ExtrCL1, 2, and 3, too. Furthermore, the same full discrimination result is present for M1_M2 class, the only difference is that the M1 and M2 film samples perform the highest proportion values (above 99.9%) and the C1, C2, C3, and C4 film samples perform proportion values between 0.265% and 23.9% (Table 7). All the previously presented facts validate the full imagistic discrimination between the film samples with extract (C1, C2, C3, and C4) and without extract (M1 and M2). Figure 5 and Figure 6 confirm the entrapment of the mucoadhesive films with extracts.

The chromatic parameters CIE L*a*b*, Yi, and Bi were analysed at pixel level. Table 7 and Figure 5 present the results in the mean with standard deviation format. The mean and standard deviation values were computed with the observation numbers from last column of Table 7. 

The chromatic parameter values were subjected to one-way ANOVA (*p* = 0.05). In order to emphasise the discrimination between the film samples with extract (C1, C2, C3, and C4) and without extract (M1 and M2), a pairwise multiple comparison with post hoc Tukey’s test (*p* = 0.05) was conducted (Table 7). Except for the lightness parameter, L*, for all other parameters a full chromatic discrimination is performed. This fact is validated by different accompanying letters for means, between the film samples with extract and without extract (Table 7). Furthermore, graphical representations from Figure 5 also validate this fact.

#### 2.3.3. Film Density

Because the films obtained were so rarefied that their density could not be estimated with the naked eye, this characteristic was determined using the optical microscope and was evaluated at 0.20 ± 0.01 mm. Three samples from different sections were taken for averaging.

The density of the films depends on the volume of solution poured into a given perimeter and the level of drying. Sodium carboxymethylcellulose-based films obtained by other researchers had density ranging from 214.8 ± 17.9–312.9 ± 22.2 µm [40] as well as 0.168 ± 0.011–0.284 ± 0.005 mm for polyvinyl alcohol-based films [38], which are close to our values. 

In a study concerning the analysis of mucoadhesive films based on 10% PVA, films with densities ranging from 0.58 ± 0.047 mm to 0.91 ± 0.070 mm were obtained, which were denser than the films prepared by us [37].

#### 2.3.4. Mass Consistency

Following the weighing we obtained an average value of 0.45 ± 0.02 g/cm^2^ film. In previous studies, for a film based on sodium carboxymethylcellulose with a diameter of 15 mm, film weights ranging from 63.2 ± 4.7 mg to 73.5 ± 5.2 mg were estimated, thus being lighter than those obtained in the present study [40]. However, films with weights close to those prepared by us were also obtained, with values ranging from 0.21 to 0.59 g/cm^2^ [39].

#### 2.3.5. Folding Endurance

The folding endurance was carried out to determine the tear strength and mechanical strength of the buccal films. After being folded 100 times in the same place, the films did not degrade. They show a high level of flexibility, which may be due to the glycerine in the composition of the films. The flexibility of the product denotes high mechanical strength. There were no differences between films with different PVA concentrations, with both films containing 5% PVA and those containing 8% PVA being equally flexible. Like other films in which PVA and glycerol were present, the fold strength was very good, and the films were very flexible [41]. Other mucoadhesive films based on hydroxypropyl methylcellulose did not break after 100 folds and had high flexibility [38].

#### 2.3.6. Tensile Strength

The results are the average of three measurements. This characteristic was measured in Newtons. Up to around 10 N the films did not break, only elongation was observed. Of the four films with plant extract, sample C2 shows the highest tensile strength, containing 5% polymer in its composition. Sample C4 has a lower tensile strength with 8% polymer. 

The polymer films obtained by us had tensile strengths similar to those obtained in a previous study, ranging from 7.07 ± 2.40–15.91 ± 3.02 N/mm^2^ for a hydroxypropylmethylcellulose-based film [36]. In testing the tensile strength of mucoadhesive films based on sodium carboxymethylcellulose (NaCMC), similar values were obtained, but also higher than ours, ranging from 17.4 ± 2.1 to 568.5 ± 62.6 N cm^−2^, these values increasing directly proportional to the increase in NaCMC concentration (Figure 7) [40].

#### 2.3.7. pH Determination

The pH values ranged from 6.634 ± 0.047 to 7.016 ± 0.026. From the control samples it can be seen that the polymer and solvent do not influence the pH value, it is neutral. The pH differences were influenced by the composition of the plant extracts, but all values were close to a neutral pH. These results illustrate the compatibility of the prepared polymer films with salivary pH (Table 8).

Saliva usually has a pH between 6.2–7.6, keeping the pH of the mouth close to neutrality [42]. The results of the MOF analysis show that they correspond to the normal pH range of the oral cavity.

Compared to other studies in which pH was determined using a pH meter, the pH of our films is close to the values obtained by other researchers, ranging between 5.5 ± 0.0 and 6.9 ± 0.0, in the case of films composed of sodium carboxymethylcellulose, glycerine, and water [40]. In the case of polyvinyl alcohol and rizatriptan-based films, the pH was determined with a pH meter and values between 6.54 ± 0.03 and 6.95 ± 0.05 were obtained, which are similar to our values and within the range of normal salivary pH [41]. In another study, the pH of mucoadhesive films was determined by applying a pH paper to the surface of the film and was between 7 and 8 [39]. 

#### 2.3.8. Disintegration Pace of In Vitro Polymeric Films

The disintegration pace is the period that a piece of a polymer film with a surface area of 1 cm^2^ takes to completely decompose in a phosphate buffer salt solution at 37 °C. According to the results, polymer films containing 5% PVA disintegrate in a shorter time than those containing 8% PVA. The disintegration time of the prepared polymer films is between 220 ± 5 s and 260 s ± 5.

Our films had a longer disintegration time, in contrast to other studies in which the films disintegrated in a range of 137 to 145 s [43]. See Table 9.

#### 2.3.9. Antioxidant Activity of Polymeric Films

The antioxidant activity of the films was determined by the DPPH approach and the results were expressed as an inhibition percentage. The mucoadhesive films with lower polymer content show higher antioxidant activity than those with higher PVA content. At the same time, MOFs with a higher amount of plant extract in their composition have higher antioxidant activity. Being a combination of extracts, the whole phytocomplex contained in the mixture is responsible for the antioxidant activity. Compared to the results obtained when analysing the corresponding plant extracts, MOFs have lower antioxidant activity due to the obviously lower extract content in a film (Table 10).

#### 2.3.10. Active Substance Content of Polymeric Films

The percentage range of active ingredients in MOF was between 36.11–88.89%, with the highest amount detected in C2 film. In other studies, this percentage was between 89.71–96%, determined by the same expression as in the current study [39]. Results are detailed in Table 11.

### 2.4. In Vitro Active Substance Release

The release of active principles from mucoadhesive films is maximal at the final disintegration phase (360 min) and is directly proportional to the diffusion pace. The values were close among the four films, ranging from 84.93% to 94.46%. The polymer matrix slowly releases the phytocomplex from the mucoadhesive film according to the kinetics study performed individually for each polymer and extract concentration.

Time series data for extract release from film samples were subjected to non-linear regression with an allosteric sigmoidal function. The R-squared values for all the analysed film samples are above 0.988 which implies high regression accuracy (Table 12). The regressions have four parameters: Vmax (%), the maximum asymptotic forecasted release value; h (a.u.), the slope of the linear part; Khalf (min), the time to achieve the half value of Vmax; Kprime (min) = Khalf ^ h.

Film sample C1 and C3 have approximate the same h-slope and Vmax values, but different Khalf values. The same behaviour is present between the C2 and C4 film samples. Furthermore, the C2 and C4 perform the Vmax asymptotic values within the 360 min, but the C1 and C3 still have a lower slope value. Figure 8 reveals the release of extract samples from the film samples.

As in other studies evaluating the diffusion of the active substance through the Franz cell, the release was directly proportional to the diffusion pace, being maximum at the final phase [44]. In a study evaluating the release of blue methylene from polymer films, samples were collected at 5, 10, 15, 30 min, 1 h, 2 h, 4 h, 6 h. The percentage of yielding was incrementing alongside the increase in time [43].

### 2.5. Evaluation of Anti-Inflammatory Effect in Acute Gingivitis

Analysing the physico-chemical and mechanical properties of the polymeric films obtained, we decided to continue the study of the anti-inflammatory effect in acute gingivitis after professional hygiene with the C2 sample. Due to the fact that the disintegration pace is slower in the case of films with 5% polymer, and the other properties studied are similar between the two PVA concentrations, we decided to continue the study with the minimum required polymer concentration. Since the films with a higher concentration of plant extract had higher antioxidant character and higher active ingredient content, we opted for the 10% plant extract concentration.

Comparing the antioxidant activity of the extracts, the tensile strength of the films, the antioxidant activity of the polymer films and the active substance content of the films, we selected the T/M 1:2 extract mixture and the film with 5% PVA and 10% extract (sample C2) for the in vivo study. Even though the release percentage of the film extract was not the highest in the C2 film, from a pharmacokinetic perspective this film consistently yields the active ingredients in the range studied.

Of the 50 patients, 30 of them were treated with mucoadhesive polymer films containing 10% Taraxaci folium/Matricariae flos plant extract mixture in a 1:2 ratio, 10 patients received placebo treatment and 10 patients were treated with a commercial reference product. The commercial reference product was a solution based on glycerine and alcoholic extracts of marigold flower, prevent root and tansy root, containing 6% (*v*/*v*) ethyl alcohol from the plant extracts.

The anti-inflammatory activity of aqueous chamomile extract and chamomile essential oil has been demonstrated on inflammation induced in laboratory animals by inhibition of prostaglandin E2 (PGE2) and nitric oxide (NO). PGE2 plays an important role in cellular inflammation, promoting capillary permeability and vasodilation. NO is a free radical involved in the inflammatory process that induces the production of pro-inflammatory cytokines. Chamomile flowers show studied anti-inflammatory effects on these inflammatory factors [45]. 

The anti-inflammatory activity of dandelion has been proven in a cell line study by inhibiting nitric oxide (NO) production and inhibiting cyclooxygenase-2 (COX2), factors responsible for the inflammatory process [5].

Patients followed the treatment for 7 days with twice-a-day application. In case of hypersensitivity to the product, the treatment is interrupted. The polymer films used were 2 cm/1 cm (L/l) in size, weighing 0.100 g/film and were individually wrapped in aluminium foil. At the end of the treatment patients completed a questionnaire whose answers were evaluated in the following charts. Patient responses were scored from 0 to 3, where 0 = very good, 1 = good, 2 = satisfactory, and 3 = unsatisfactory. The statistical interpretation of the results is detailed in Figure 9, Figure 10, Figure 11, Figure 12, Figure 13, Figure 14, Figure 15, Figure 16, Figure 17 and Figure 18.

## 3. Materials and Methods

### 3.1. Materials

The reagents used in this study were of analytical purity. Ethyl alcohol 96°, polyvinyl alcohol (PVA), glycerol were purchased from Nordische Oelwerke Walther Carroux GmbH & Co. KG, Hamburg, Germany, saline phosphate buffer solution (PBS), 1,1-diphenyl-2-picrylhydrazyl (DPPH), 6-hydroxy-2,5,7,8-tetramethylchroman-2-carboxylic acid (Trolox), Folin–Ciocâlteu reagent, methyl alcohol, acetic acid (≥99%), sodium carbonate, acetonitrile, aluminium chloride (≥98%), sodium nitrite, sodium hydroxide, copper chloride, neocuprein, ammonium acetate buffer were purchased from Sigma-Aldrich, Budapest, Hungary.

The standards used for chromatographic analysis of sterols were ergosterol (≥95%), campesterol (~65%), stigmasterol (~95%), purchased from Sigma-Aldrich, Schnelldorf, Germany and beta-sitosterol (≥80%) purchased from Carl Roth, Karlsruhe, Germany. 

Reference standards for α-tocopherol (≥95.5%) and γ-tocopherol (≥96%) were procured from Sigma-Aldrich, Schnelldorf, Germany and δ-tocopherol was purchased from Supelco, Bellefonte, PA, USA. 

Eupatorin, casticin, and hispidulin standards from Sigma-Aldrich, Schnelldorf, Germany were used for the analysis of methoxylated flavones.

The standards used in the chromatographic analysis of polyphenols were as follows: caftaric acid (≥97%), caffeic acid (≥98%), chlorogenic acid (≥95%), p-coumaric acid (≥98%), isoquercitrin (≥98%), rutozide (≥97%), quercitrin (≥78%), patuletin (≥98%), luteolin (≥98), kaempferol (≥97%), apigenin (≥95%), syringic acid (≥95%), vanillic acid (≥97%) and protocatechuic acid (≥97%) purchased from Sigma-Aldrich; ferulic acid (≥99%) and quercetol from Merck, Darmstadt, Germany.

Apparatus and utensils used were: Franz Microette Hanson diffusion system, model 57-6AS9, USA, Shimadzu Spectrophotometer UV spectrophotometer, Tokyo, Japan, Sension™ 1 digital pH meter, Hach Company, Loveland, CO, USA, Brookfield CT3 Brookfield texture analysis instrument, Middleboro, MA, USA, Optika B-290 series optical microscope, Ponteranica Italy, Kern ABS 220-4N analytical balance, Kern, Lohmar, Germany, magnetic stirrer with heating function RCT 5, IKA^®^, Staufen, Germany, electronic pipettes, telephone—stopwatch, Berzelius beakers, Petri dishes, watch glass, Pasteur pipettes, spectrophotometer cuvettes, magnets, aluminium foil, linear, scalpel, gloves, disinfectant, scissors, tweezers, graduated flasks, test tubes, spatulas, gauze, spectrophotometer cuvettes, gloves, disinfectant.

An HPLC 1100 series system from Agilent Technologies, Santa Clara, California, USA, equipped with desagglomerator, binary gradient pump, thermostat, UV detector and Zorbax SB—C18 analytical column (100 × 3.0 mm i.d. with 3.5 µm particles) was used for the chromatographic analyses. The HPLC system was coupled with an Agilent Ion Trap 1100 SL mass spectrometer. Brucker Ion Trap SL from Brucker Daltonics GmbH, Leipzig, Germany.

Chromatographic data were processed with ChemStation (vA09.03) and DataAnalysis (v5.3) software from Agilent, USA. Antioxidant analysis data were processed with Microsoft Excel. 

### 3.2. Plant Extract

The plant material used, the chamomile flowers (Matricariae flos—M) and dandelion leaves (Taraxaci folium—T) was obtained from medicinal teas, purchased from Fares, Orăștie, Romania. Extracts of 10% (*w*/*w*) concentration were obtained from the plant product by maceration with 70% (*v*/*v*) ethyl alcohol. The plant product was left to macerate for 10 days in a cool room protected from light, then filtered and the solution was stored at 6 °C (±2 °C). After 5 days the plant extracts were decanted and mixtures were obtained in the ratio of 1:1 MT, 1:2 MT and 2:1 MT. The extracts were stored in brown glass containers at 6 °C (±2 °C). 

### 3.3. Identification and Quantification of Active Ingredients in Plant Extracts

#### 3.3.1. Determination of Total Flavonoid Content

The samples to be analysed were obtained as follows: in a 10 mL volumetric flask, we introduced 1 mL of extractive solution mixed with 4 mL deionised water and then added 0.3 mL 5% sodium nitrite solution. After 5 min we added 0.3 mL 10% aluminium chloride solution, and after 6 min 2 mL of 1 M sodium hydroxide solution. We filled the volumetric flask to the mark with deionised water and shook it [46].

Determination of quercetin content was performed through spectrophotometric method, reading extinction at 510 nm wavelength. The control sample was composed of deionised water, 0.3 mL 5% sodium nitrite solution, 0.3 mL 10% aluminium chloride solution and 2 mL sodium hydroxide solution. The calibration curve for the determinations was drawn using the quercetin standard. The flavonoid content of the extracts was calculated from the regression equation derived from the graph and expressed as mg quercetin equivalents (QE)/100 g dry extract, where y = 0.8388x + 0.0003 and the correlation coefficient is R^2^ = 0.9966.

#### 3.3.2. Determination of Total Polyphenol Content

We determined the total polyphenol content through the Folin–Ciocâlteu assay. The presence of polyphenols in the extracts was revealed by means of the blue compounds formed between the hydroxyl groups of the sample and the Folin–Ciocâlteu reagent in alkaline medium adjusted with sodium carbonate.

The samples were obtained from 0.1 mL alcoholic extract mixed with 1.7 mL deionised water and 0.2 mL Folin–Ciocâlteu reagent, gently shaken, and after 5 min we added 1 mL 20% sodium carbonate solution to obtain a pH = 10. The basic pH is necessary for the reaction between polyphenols and Folin–Ciocâlteu reagent. The solution tubes were shaken and left in the dark for 90 min [47].

At 765 nm wavelength, the optical denitrification increases in proportion to the number of hydroxyl groups in the polyphenol (anthocyanin) structure. The absorbance of the blue solutions was measured spectrophotometrically at 765 nm against the control sample, consisting of deionised water, 0.2 mL Folin–Ciocâlteu reagent and 1 mL 20% sodium carbonate solution. The calibration curve was performed with gallic acid solutions of 20–100 ppm. The total polyphenol content of the extracts was calculated from the regression equation y = 0.0135x + 0.0832 and expressed as mg gallic acid equivalents (GAE)/100 g dry extract.

#### 3.3.3. Identification and Quantification of Sterolic Compounds

High-performance liquid chromatography coupled with mass spectrometry (HPLC/MS) was used for the analysis of sterols in pure plant extracts.

Phytosterol compounds were separated using a Zorbax SB-C18 reversed-phase analytical column (100 mm × 3.0 mm i.d., 5 µm particles) equipped with a Zorbax SB-C18 guard column operated at 40 °C. Sterol compounds were separated by isocratic elution using a mixture of methanol:acetonitrile 10:90 (*v*/*v*) as mobile phase. It was operated at a temperature of 45 °C, with a flow rate of 1 mL/min and injection volume of 5 µL. An Agilent Ion Trap 1100 SL mass spectrometer with atmospheric pressure of chemical ionisation interface (APCI) was used for mass spectrometry analysis. The ionisation status was positive. To achieve maximum sensitivity values, conditions were optimised. It was created in nitrogen medium with nebulisation pressure of 60 psi and vaporisation at 400 °C. The temperature of the drying gas (nitrogen) was 325 °C with a flow rate of 7 L/min. The capillary potential was −4000 V [48].

Four standards were used for quantitative determination: beta-sitosterol, stigmasterol, campesterol and ergosterol. Under the chromatographic conditions presented, the retention times for the four sterols analysed were: 2.4 min for ergosterol, 3.7 min for stigmasterol and campesterol (coelution) and 4.2 min for beta-sitosterol [49].

Agilent ChemStation (vA09.03) and DataAnalysis (v5.3) software were used to obtain and analyse chromatographic data.

#### 3.3.4. Identification and Quantification of Tocopherols

The alpha-tocopherol, gamma-tocopherol and delta-tocopherol content of the plant matrix was determined by a validated high-performance liquid chromatography-mass spectrometry method. This method was a rapid method with a run time of only 6 min and good resolution. The HPLC system used was an Agilent 1100 series (binary pump, autosampler, thermostat—Agilent Technologies, Santa Clara, CA, USA), coupled with a Brucker Ion Trap SL (Brucker Daltonics GmbH, Leipzig, Germany). A Zorbax SB-C18 chromatography column (100 × 3.0 mm and i.d., 3.5 µm—Agilent Technologies) was used [50].

Stock standard solutions of tocopherols were prepared in methanol, concentration of 1 mg/mL. From these, working solutions were obtained by dilution in water/acetone 50:50 (*v/v*). Retention times were 3.3 min for delta-tocopherol, 4.1 min for gamma-tocopherol, and 5.1 min for alpha-tocopherol, respectively. A volume of 10 µL plant extract was injected into the liquid chromatography/mass spectrometry system. Calibration curves for the three tocopherols were linear in the concentration range from 40 ng/mL to 960 ng/mL, with a correlation coefficient R2 greater than 0.99. Results were expressed in ng tocopherol/mL plant extract.

#### 3.3.5. Identification and Quantification of Methoxylated Flavones

High-performance liquid chromatography coupled with mass spectrometry (HPLC/MS) was used for the analysis of methoxylated flavone aglycones in plant extracts.

HPLC-MS analysis of methoxylated flavones was performed using an Agilent 1100 series HPLC chromatography system with binary pump, autosampler and thermostat, coupled with an Agilent 1100 Ion Trap 1100 SL mass spectrometer. Separation was performed on a Zorbax SB-C18 reverse-phase analytical column (100 × 3.0 mm i.d., 5 µm particle size, using linear gradient elution with mobile phase consisting of 0.1% acetic acid:methanol *v/v*, starting with 45% methanol and ending with 50% methanol over 8 min. Working temperature was 48 °C, flow rate 0.9 mL/min, and injection volume 5 µL [51].

Detection was performed in MS/MS in an MRM (multiple reaction monitoring) analysis mode using electrospray ionisation (ESI) as the ionisation source in a negative mode. It was worked in nitrogen medium with a nebulisation pressure of 60 psi. The temperature of the drying gas (nitrogen) was 325 °C, with a flow rate of 12 L/min. The capillary potential was +2500 V. 

Three standards were used for quantitative determination: hispidulin, eupatorin and casticin. Detection was performed by monitoring the specific ions of the tested flavones. Under the previous chromatographic conditions, the retention times of the six flavones analysed are: 4.2 min for hispidulin, 7.6 min for eupatorin and 8.05 min for casticin. Complete identification of the compounds was achieved by comparing retention times and mass spectra with those of the standards under the same chromatographic conditions. Results were expressed as µg methoxylated flavone/mL extract.

#### 3.3.6. Identification and Quantification of Polyphenolic Compounds

Polyphenolic compounds were analysed through two methods.

In the first method, a series of polyphenols were qualitatively and quantitatively identified using the following standards: caftaric acid, gentisic acid, caffeic acid, chlorogenic acid, p-coumaric acid, ferulic acid, isoquercitrin, rutozide, quercitrin, quercetol, patuletin, luteolin, kaempferol, and apigenin.

Polyphenol identification was performed using an Agilent HPLC 1100 Series system equipped with degasser, binary gradient pump, column thermostat, autosampler and UV detector. The HPLC system was coupled with an Agilent 1100 mass spectrometer (LC/MSD Ion Trap SL, Santa Clara, CA, USA). A reverse phase analytical column (Zorbax SB-C18 100 × 3.0 mm i.d., 3.5 µm particles) was used for separation and operated at 48 °C. Detection of compounds was performed in both UV and MS modes.

The UV detector was set at 330 nm until 17.5 min, then at 370 nm until the end of the analysis time. The MS system was operated using an electrospray ion source in negative status. The mobile phase was a binary mixture prepared from methanol and 0.1% *v*/*v* acetic acid solution. The elution started linearly with 5% methanol and ended with 42% methanol for 35 min. The isocratic elution was carried out over the next 3 min with 42% methanol. Flow rate was 1 mL/min and injection volume was 5 µL [52].

Signal MS was used for qualitative analysis only, polyphenols were identified based on their specific mass spectra. MS spectra obtained from a standard solution of polyphenols were integrated into a mass spectral library. Subsequently, the MS traces/spectra of the analysed samples were compared with the spectra in the library, allowing positive identification of compounds based on spectrum matching. The UV trace was used to quantify the compounds identified from the MS detection [52].

Using the chromatographic conditions described previously, the polyphenols eluted in less than 35 min. 

Four polyphenols cannot be quantified under current chromatographic conditions due to overlap (caftaric acid with gentisic acid and caffeic acid with chlorogenic acid). However, all four compounds can be selectively identified in MS detection (qualitative analysis) based on differences in their molecular mass and MS spectra.

The detection limits were calculated as a minimum concentration, producing a reproducible peak with a signal-to-noise ratio greater than three. Quantitative determinations were performed using an external standard method. The limit of quantification for this method was 0.5 µg/mL and the limit of detection was 0.1 µg/mL. Results were expressed in µg polyphenol/mL extract. Data were processed with Agilent’s ChemStation (vA09.03) and DataAnalysis (v5.3) software (Santa Clara, CA, USA).

For the second method for the identification of polyphenols in plant extracts, an LC-MS method was performed using three polyphenols as standards, that is syringic acid, vanillic acid and protocatechuic acid. 

Analytical separation was performed on Zorbax SB-C18 analytical column, 100 mm × 3.0 mm i.d., with 3.5 µm particles, with a 0.1% (*v*/*v*) methanol/acetic acid mixture as mobile phase and binary gradient, starting with 3% methanol, continuing 3 min with 8% methanol, reaching 8.5 min with 20% methanol up to 10 min, and then re-equilibrating the column with 3% methanol. The injection volume was 5 µL with a flow rate of 1 mL/min. Detection of compounds was performed in MS mode (SIM-MS). The MS system was operated using an electrospray ion source in negative status. It was operated in nitrogen medium with 60 psi nebulisation pressure and +3000 V capillary potential. The temperature of the drying gas (nitrogen) was 360 °C and the flow rate was 12 L/min [53]. The results were expressed in µg polyphenol/mL extract.

HPLC plots for the standard sample used for identification and quantification of polyphenolic compounds and for the evaluated samples are provided in the Appendix A.

### 3.4. Determination of the Antioxidant Capacity of Plant Extracts

#### 3.4.1. Cuprac Assay—Reduction of Cu^2+^ Copper Ions

The antioxidant capacity of extracts against copper ions (Cu^2+^) was determined through the Cuprac method. This experiment is based on the changes in the absorption characteristics of the Neocuprein (Nc) complex when reduced by an antioxidant. The reduction potential of the sample consists in the conversion of Cu^2+^ ions to Cu^1+^ ions, through a redox reaction.

We obtained samples from 0.5 mL 10 mM copper(II) chloride solution, 0.5 mL 7.5 mM neocuprein ethanolic solution and 0.5 mL 1 M ammonium acetate buffer solution (to maintain pH), then added 0.49 mL deionised water and 0.01 mL alcoholic extract. The tubes were closed and kept at room temperature for 30 min [54].

Absorbances were read at a wavelength of 450 nm relative to the control sample, obtained from the solvents used to prepare the samples, in the appropriate amounts. A high optical density of the samples denotes an increased reducing capacity of the extracts. The calibration curve was performed using Trolox between concentrations of 0 and 2500 µM, with correlation coefficient R2 = 0.9935. The antioxidant capacity of the extracts was calculated using the regression equation y = 0.0006x, where x represents Trolox equivalent µmol in 100 µL extract and y represents the absorbance read at 450 nm.

#### 3.4.2. DPPH Assay—Determination of Antioxidant Capacity

In vitro antioxidant capacity assessment was performed through DPPH technique using 1,1-diphenyl-2-picrylhydrazyl (DPPH) reagent. Screening and reduction of free radicals was carried out through DPPH method. We determined the ability of the extracts to reduce the compound 1,1-diphenyl-2- picrylhydrazyl by a colourimetric method. DPPH can react with an antioxidant substance that can donate hydrogen and reduce DPPH. With this method, we determined the antioxidant capacity by testing the ability of the extracts to neutralise free radicals. The DPPH method is often used to quantify antioxidants in complex biological sites.

DPPH is a purple compound, stable at room temperature and showing characteristic absorption at 517 nm wavelength. The free nitrogen radical of the DPPH reagent is readily reduced by an antioxidant complex to a yellow complex, namely 1,1-diphenyl-2-picryl-hydrazine. The DPPH solution in 6 × 10^−5^ ethanol was freshly prepared before spectrophotometric determinations were carried out. From this solution, we took 2.9 mL, which we mixed with 0.1 mL alcohol extract and 2 mL distilled water. The samples were left in the dark for 30 min, after which we recorded the absorbances at 517 nm relative to the control sample. The control sample was the DPPH solution [55].

The reduction of free radicals is observed by the change in colour of the solutions from dark purple to light yellow. The discolouration is stoichiometric with the number of electrons gained. 

The calibration curve was performed using Trolox between known concentrations. The correlation coefficient is R^2^ = 0.9674. The percentage inhibition of DPPH was calculated using the equation below: [55]
$$\%Inhibition=\frac{\left(A_{Blank}-A_{Sample}\right)}{A_{Blank}}\times 100$$

#### 3.4.3. Statistical Analysis

Data were statistically analysed using the one-way analysis of variance (ANOVA) with a Tukey post hoc test. Data analysis was performed with Prism software, version 9.3.0 (GraphPad Software, Boston, MA, USA). Results were considered statistically significant when p values were below 0.05.

### 3.5. Formulation of Polymeric Films

The formulation of mucoadhesive polymer films is predominantly achieved by solvent casting. The first step involves dispersing or dissolving the polymer in a solvent or mixture of solvents, the most commonly used being water and alcohol. Plasticisers, penetration enhancers and fillers may be added to this mixture. Various active pharmaceutical ingredients, dissolved or suspended, are introduced into the casting solution. The resulting viscous solution is poured into moulds and allowed to dry, then packaged [56].

The pharmaceutical form devised was of the matrix type. Its configuration comprises the active substance, the polymer and the excipients mixed together into a unitary product [44].

Out of the water-soluble polymers, we chose to use polyvinyl alcohol for the preparation of MOF. The solvent used to disperse the polymer was composed of glycerine, 96° ethyl alcohol and distilled water in a ratio of 1:4:5.

Following the antioxidant activity, the mixture with the ratio Taraxaci folium/Matricariae flos 1:2 was selected.

In a Berzelius beaker, the freshly prepared solvent and magnet were placed on the magnetic stirrer. The polyvinyl alcohol is dispersed in the solvent under stirring. The solution is stirred for one hour at 50 °C, then stirred for another 6 h at full speed at room temperature to obtain a homogeneous solution. After this time the plant extract is added and stirred for another hour at low speed. A control sample was prepared for each polymer concentration.

The solution is poured into glass jars and left to dry for 48 h in a cool dry place. After this time the film is removed from the mould, weighed and cut to a convenient size. The polymer films were packed in aluminium foil. The polymer films were prepared under aseptic conditions.

Polymer films were prepared in 5% and 8% polyvinyl alcohol concentrations and 5% and 10% plant extract were used for each polymer concentration, respectively. Four different concentrations of polymer films were thus obtained, noted according to the table below. For each polymer concentration, a control sample was obtained without plant extract (Table 13).

### 3.6. Evaluation of Polymeric Films Characteristics

#### 3.6.1. Appearance

The appearance of the polymeric films was determined through evaluation of organoleptic properties and using the optical microscope, and the sensory analysis was carried out using the Hedonic test.

The Hedonic test is an acceptability test based on the sensory analysis of a product using a series of questions and has a high degree of subjectivity. The participants can be either trained or untrained tasters (consumers). At least 50 participants are required for the test to be valid. The samples evaluated must be identical in appearance. The taste of the product was rated on a scale from 1 to 9, where 1 = extremely unpleasant and 9 = extremely pleasant [57].

Before each tasting, participants cleaned their oral cavities with water. The sample was allowed to dissolve slowly in the oral cavity without being swallowed.

#### 3.6.2. Film Samples Chromatic and Imagistic Analysis

The film samples were scanned with Canon CanoScan 9000F optical scanner (Canon Inc., 30-2, Shimomaruko 3-chome, Ohta-ku, Tokyo, Japan). Image pixel resolution was 600 dpi (0.0423 mm) and the pixel colour depth was 48-bit. Before each sample scanning, a colour self-calibration was performed. In order to cancel embedding noise, the scanned images were pre-processed with Corel PaintShop Pro v2012 v23.1.0.27x64 (Corel Alludo HQ, 333 Preston Street, Suite 700, Ottawa, ON, Canada). The chromatic parameters: CIE L*a*b*, Yellow index (Yi) and Browning index (Bi) were calculated with a custom-made software with MATLAB R2023a v9.14.0 64 bit CWL (MathWorks, 1 Apple Hill Drive Natick, MA, USA). Furthermore, in order to assess the imagistic parameters were proposed six chromatic classes: ExtrCL1, ExtrCL2, ExtrCL3, ExtrCL4 (for film samples with extract), M1_M2 (for film samples without extract) and Artefact (for some artefacts inside the films). A histogram-based image segmentation algorithm was performed, and the results reflect the proportion of pixels that meet the chromatic criteria of each chromatic class.

The univariate statistic test, one-way ANOVA (*p* = 0.05), combined with post hoc Tukey’s multiple comparisons test (*p* = 0.05) was used to compare the film samples images. Univariate statistical analysis was performed with Stata 17SE statistical v.6 software (StataCorp LLC, 4905 Lakeway Drive, College Station, TX, USA).

#### 3.6.3. Density

The density of the films was analysed under an optical microscope. Three samples of each film were measured using the graduations on the microscope slide, and then the measurements were averaged [40].

#### 3.6.4. Mass Consistency

For each film type, ten samples of 1 cm^2^ size from different sections were picked and weighed on the analytical balance. To calculate the mass consistency the average of the three weightings was calculated [38].

#### 3.6.5. Folding Endurance

A film of each concentration, 1 cm^2^ in size, was selected and folded 100 times through the same place. The number of folds characterizes the flexibility of the polymeric film [38].

#### 3.6.6. Tensile Strength

The tensile strength and rupture characteristics of the polymeric films were evaluated using the Brookfield CT3 texture analysis tool. It contains two load cell handles, of which the lower one is fixed and the upper one is movable. The analysed films were cut to the size of 1 × 0.5 cm, clamped between the cell handles, and force was applied incrementally until the film broke. The tensile strength was measured in N [38].

#### 3.6.7. pH Determination

The polymeric films were dissolved in bidistilled water whose pH was previously determined and found to have a neutral value. The pH of the polymeric films was determined using a pH meter [41].

#### 3.6.8. In Vitro Disintegration Time of Polymeric Films

Phosphate buffer saline was prepared in a 1 L volumetric flask by diluting PBS 10 times. By evaluating the disintegration in phosphate buffer saline, an analogy is created with the pH of saliva, thus considering the PBS solution as artificial saliva. In a Berzelius beaker, 25 mL of freshly prepared phosphate buffer heated to 37 °C is placed on the magnetic stirrer. A polymer film with a surface area of 1 cm^2^ is placed in the beaker and the time required for disaggregation is timed [43].

#### 3.6.9. Antioxidant Activity of Polymeric Films

Polymeric films were dissolved in 10 mL of freshly prepared PBS buffer. The samples taken in the study were 1 cm^2^ in size. After complete dissolution, 0.5 mL solution was taken and processed by DPPH assay to determine the antioxidant character of the polymer films.

The evaluation of the in vitro antioxidant capacity was carried out by DPPH technique using 1,1-diphenyl-2-picrylhydrazyl (DPPH) reagent. The DPPH solution in 6 × 10^−5^ ethanol was freshly prepared before spectrophotometric determinations. To 2.9 mL DPPH solution was added 0.1 mL polymer film solution and 2 mL distilled water. Absorbances were read at 517 nm after the samples had been left in the dark for 30 min. The percentage inhibition was calculated using the equation below [55]:$$\%Inhibition=\frac{\left(A_{Blank}-A_{Sample}\right)}{A_{Blank}}\times 100$$

#### 3.6.10. Active Ingredient Content of Polymeric Films

To estimate the content of active principles in polymer films, 1 cm^2^ of each polymer film was used and dissolved in 10 mL PBS solution. After complete dissolution of the films, samples were taken, and absorbance read at 370 nm. The active ingredient content was expressed in mmol/L quercetin using the calibration curve. Values were expressed as a percentage using the equation below [39]:$$\%=\frac{A_{Sample}}{A_{Blank}}\times 100$$

#### 3.6.11. In Vitro Active Ingredient Release from Polymeric Films

In vitro release was carried out using a Franz cell system with synthetic membranes, using a 6-cell diffusion system with a diffusion area of 1.767 cm^2^ and a volume of 6.5 mL for the receptor chamber. The receptor chamber in each cell was filled with phosphate buffer saline mixed with freshly prepared 30% ethanol. Synthetic membranes were hydrated by immersion in the receptor medium for 30 min before use, and then fitted between the donor and acceptor compartments of the Franz diffusion cell. Approximately 0.100 g sample was weighed and then applied to the membrane surface. The diffusion cells were tightly closed by clamping. The system was maintained at 32 °C with continuous stirring at 600 rpm. 0.5 mL of receptor solution was withdrawn at 5, 10, 15, 30, 60, 120, 180, 240, 300, and 360 min and replaced with fresh receptor medium to maintain constant volume (6.5 mL) throughout the assay period. Samples were analysed in a UV–VIS spectrophotometer at 370 nm. Each film was tested in triplicate [44].

The film extract release data were subjected to non-linear regression with allosteric sigmoidal function. This statistical analysis was performed with GraphPad Prism v5.3 (GraphPad Software, 225 Franklin Street. Fl. 26, Boston, MA, USA).

### 3.7. Evaluation of the Anti-Inflammatory Activity of Polymeric Films in the Treatment of Acute Gingivitis

In the human subjects study, 50 patients diagnosed with gingivitis after plaque removal were selected.

In choosing the number of patients, we also referred to other studies in which mucoadhesive oral films were studied in vivo and in which the number of patients was smaller than in the current study, being less than 15 participants. In addition, because it is a new product for dental practice, because people are reluctant to participate in such studies and because they are still careless with oral hygiene, we chose to work with a number of 50 patients [58,59,60].

The inclusion criteria for the study were: patients who have signed informed consent, patients over 18 years of age, females and males, patients diagnosed with acute gingivitis after professional hygiene, and clinically healthy patients.

Exclusion criteria from the study were: non-compliant patients, pregnant women, patients under 18 years of age, and patients with general or localised conditions.

All patients signed an informed consent to participate in the study. Patients received non-invasive treatment which they followed at home and were examined periodically. Patients were not exposed to any psychological risk. The aim of the study was to accelerate gingivitis healing, relieve symptoms and improve the patient’s quality of life. For the clinical study the consent of the Research Ethics Committee of the Faculty of Medicine and Pharmacy of Oradea, University of Oradea, No. CEFMF/01 of 30.06.2022 was obtained.

Patients were divided into 3 groups. The first batch consisted of 30 patients, and they received treatment with the herbal film. The second group included 10 patients who were treated with the placebo film. The third group included 10 patients who followed treatment with a commercial reference product, which was a solution based on glycerine and alcoholic extracts of marigold flower, prevent root and tansy root, containing 6% (*v*/*v*) ethyl alcohol from the plant extracts.

The selected patients underwent a professional dental hygiene procedure (plaque removal and professional brushing). Patients were instructed on proper oral hygiene and advised not to use any other oral hygiene active product in addition to brushing. Some patients received mucoadhesive polymer films containing plant extract, some received mucoadhesive polymer films without plant extract, and some received a commercial reference product. Patients are advised to use the polymer film in the lingual area of the lower front teeth in the canine-canine region. No food or liquid should be consumed for half an hour after application of the product. At the end of the treatment, the patient completed a questionnaire on the efficacy of the studied product. Patient responses were scored from 0 to 3, where 0 = very good, 1 = good, 2 = satisfactory, and 3 = unsatisfactory.

Patients were screened by examining the gingival index (GI), which quantifies the status/severity of gingival inflammation. GI was assessed by inserting a periodontal probe into the gingival sulcus or gingival recesses. GI scores are:

0—normal gum with no signs of inflammation and no bleeding;

1—mild inflammation, slight discolouration, no bleeding;

2—moderate inflammation, erythema, oedema, bleeding on probing or pressure;

3—severe inflammation, pronounced redness, oedema, spontaneous bleeding and sometimes ulceration [61].

Gingival index was calculated after professional hygiene, at 48 h and 7 days.

## 4. Conclusions

According to the phytochemical profile examined, the two extracts studied combine harmoniously to form a complex source of polyphenols with antioxidant and anti-inflammatory activity. The sterols analysed in this study were present in both Matricariae flos extract and Taraxaci folium extract. Of the three tocopherols determined, only alpha- and gamma-tocopherols were common to both extracts, but in higher amounts in the extract obtained from chamomile flowers. Methoxylated flavones were predominant in the Matricariae flos extract, as were phenolic acids. Of the remaining polyphenols quantified in both extracts, chlorogenic acid was identified in higher amounts and luteolin in much lower amounts. By determining the total flavonoid and polyphenol content, the values obtained were higher in the Taraxaci folium extract, and among the three mixtures obtained the differences were small, but slightly higher for the Taraxaci folium/matricariae flos 1:1 (*m*/*m*) mixture. The antioxidant character determined in the individual extracts was more pronounced for the Matricariae flos extract. Through the DPPH method the antioxidant character was more pronounced in the Taraxaci folium/matricariae flos 1:2 (*m*/*m*) mixture and through the Cuprac method the Taraxaci folium/matricariae flos 2:1 (*m*/*m*) mixture was found to have a slightly higher value than the Taraxaci folium/matricariae flos 1:2 (*m*/*m*) mixture.

The antioxidant activity of individual and combined extracts was determined by DPPH and Cuprac methods. For individual extracts, the antioxidant capacity was higher for chamomile extract by both methods performed. By the DPPH method, the antioxidant capacity was more pronounced in the Taraxaci folium/matricariae flos 1:2 (*m*/*m*) mixture, the percentage of inhibition being higher than in Taraxaci folium extract and lower than in Matricariae flos extract. By the Cuprac assay, it was found that the Taraxaci folium/matricariae flos 2:1 (*m*/*m*) mixture had a slightly higher value than the other two mixtures, but was very close to the Taraxaci folium/matricariae flos 1:2 (*m*/*m*) combination. By the method mentioned above, the plant mixtures showed higher values for the dandelion extract, but lower for the chamomile extract.

Based on the previous results, the Taraxaci folium/matricariae flos 1:2 (*m*/*m*) extract mixture was chosen for the preparation of mucoadhesive polymer films. Following the study of physicochemical and mechanical properties, the obtained mucoadhesive polymer films show good characteristics for their use in the treatment of acute gingivitis. Initially, four types of plant extract films were obtained, rated from C1 to C4 according to the percentage of polymer and extract in their composition. According to organoleptic analysis, they show characteristics suitable for oral administration, being aesthetic, odourless, and pleasant to taste and touch. The pH of the films was within the range of normal salivary pH values. The mechanical properties were appreciable for all films but with a higher value for the C2 film (8% PVA and 10% extract) regarding tensile strength. The disintegration of the polymer films proceeded more rapidly in the case of 5% PVA concentrations compared to 8% PVA. Antioxidant capacity and active ingredient content were higher for films with 10% extract in the composition than for those with 5% extract. The in vitro release of active ingredients from the polymer film through the Franz diffusion system proceeded with increasing release time, obtaining the highest final value for the 5% polymer and 5% extract film.

C2 film containing 5% PVA and 10% plant extract was chosen to study the anti-inflammatory effect in vivo. The results obtained were appraised, the usefulness of the product was evaluated as very good, and the patients were satisfied with the treatment. The mucoadhesive films studied were very easy to apply and had no adverse reactions.

Because there are no studies in the literature using the Taraxaci folium/matricariae flos extract mixture for pharmacological activity in oral diseases, and because there are no mucoadhesive films in dental practice to be used for the treatment of gingival inflammation, the formulation we introduce represents a novelty in the treatment of acute gingivitis.

## Figures and Tables

**Figure 1 molecules-28-04002-f001:**
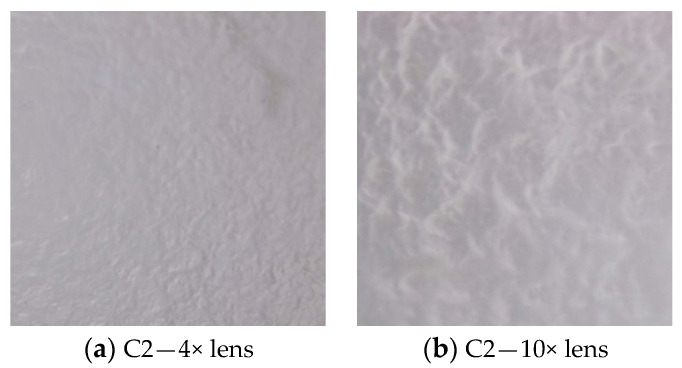
Microscopic images of polymer films. According to the optical microscope images, the appearance of the polymer films can be seen both through the 4× optical objective (**a**) and the 10× optical objective (**b**). They have a homogeneous display, from which it can be seen that the polymer particles have been uniformly distributed in the film-forming matrix.

**Figure 2 molecules-28-04002-f002:**
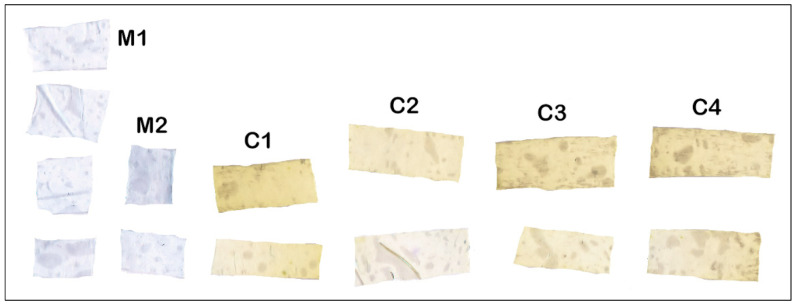
All film samples image, with and without extract. M1—5% PVA, M2—8% PVA, C1—5% PVA 5% extract, C2—5% PVA 10% extract, C3—8% PVA 5% extract, C4—8% PVA 10% extract.

**Figure 3 molecules-28-04002-f003:**
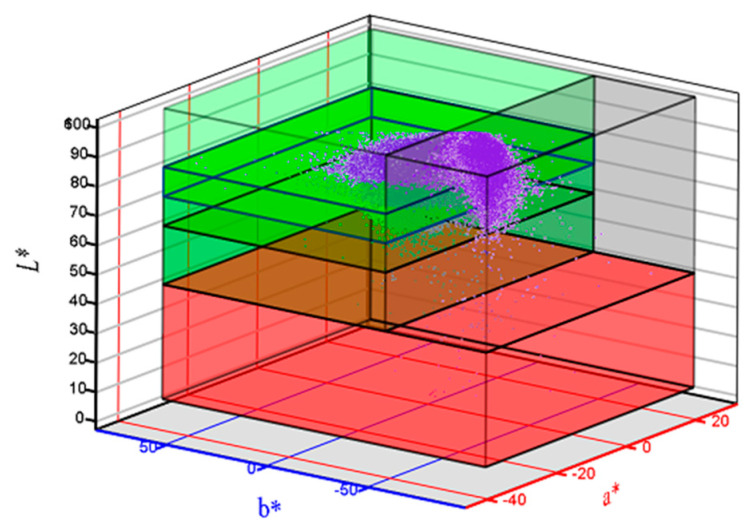
Point cloud from all samples image and the chromatic segmentation criteria represented as boxes—CIE L*a*b* chromatic space representation.

**Figure 4 molecules-28-04002-f004:**
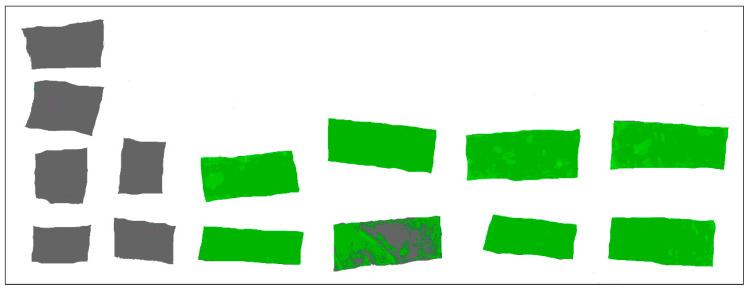
All film samples classified image, with and without extract, according to Figure 2.

**Figure 5 molecules-28-04002-f005:**
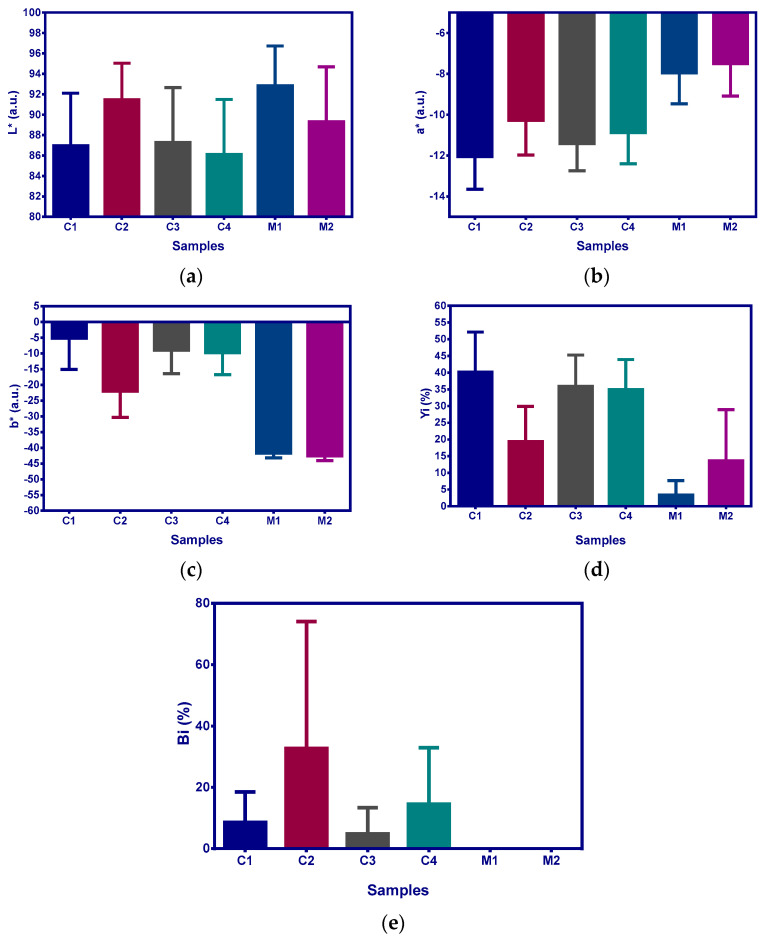
The chromatic parameters of film samples—data represented as mean values with standard deviation error lines. M1—5% PVA, M2—8% PVA, C1—5% PVA 5% extract, C2—5% PVA 10% extract, C3—8% PVA 5% extract, C4—8% PVA 10% extract. (**a**) represents the interpretation of the values for the chromatic parameter L*. (**b**) represents the interpretation of the values for the chromatic parameter a*. (**c**) represents the interpretation of the values for the chromatic parameter c*. (**d**) represents the interpretation of the values for the chromatic parameter Yi. (**e**) represents the interpretation of the values for the chromatic parameter Bi. By chromatic parameters we appreciated the enrichment of the films with extract, the C2 film having better results.

**Figure 6 molecules-28-04002-f006:**
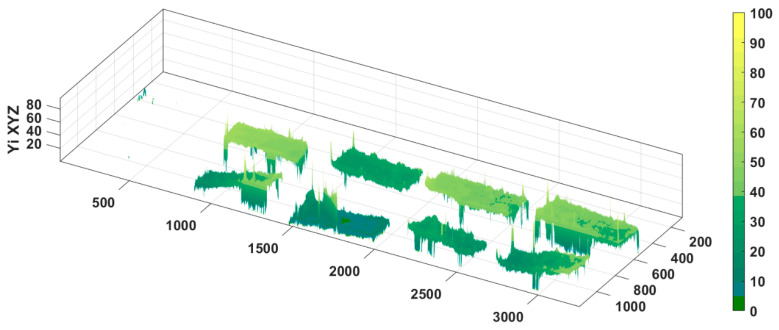
Yellow index spatial distribution of all film samples image, with and without extract. The 3D representation of the yellow index (Yi) values for image from Figure 2, facilitate a better visual understanding of the full imagistic and chromatic discrimination of the film samples with extract (C1, C2, C3, and C4) and without extract (M1 and M2). M1—5% PVA, M2—8% PVA, C1—5% PVA 5% extract, C2—5% PVA 10% extract, C3—8% PVA 5% extract, C4—8% PVA 10% extract.

**Figure 7 molecules-28-04002-f007:**
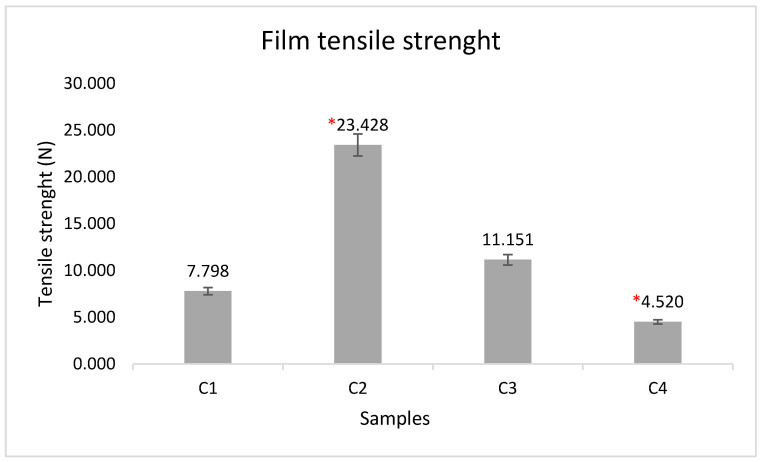
Tensile strength of mucoadhesive films. Tensile strength was measured in N, each result representing the mean ± SD, n = 3. The values obtained ranged from 4.520 ± 0.226 to 23.428 ± 1.171 N. The data were analysed with Kruskal–Wallis test followed by Dunn’s multiple comparisons test. Significant differences in the formulations were detected between the compositions. Significant differences are marked on the figure with * (*p* < 0.05), showing the significance levels in the case of composition C2 and C4. M1—5% PVA, M2—8% PVA, C1—5% PVA 5% extract, C2—5% PVA 10% extract, C3—8% PVA 5% extract, C4—8% PVA 10% extract.

**Figure 8 molecules-28-04002-f008:**
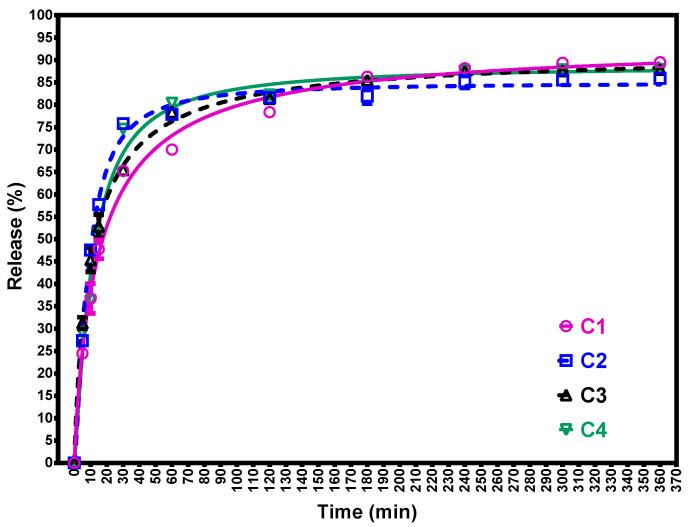
Film sample extract release. M1—5% PVA, M2—8% PVA, C1—5% PVA 5% extract, C2—5% PVA 10% extract, C3—8% PVA 5% extract, C4—8% PVA 10% extract. The results obtained from the release of the active ingredients were expressed as percentages representing the mean ± SD, n = 3. Time of release was curved between 0 and 360 min, with increasing release starting from minute 5. The highest percentage of release was obtained for the film C1, and the lowest percentage of release was obtained for the film C2. C2 film consistently releases the amount of active ingredients over the study period of 360 min.

**Figure 9 molecules-28-04002-f009:**
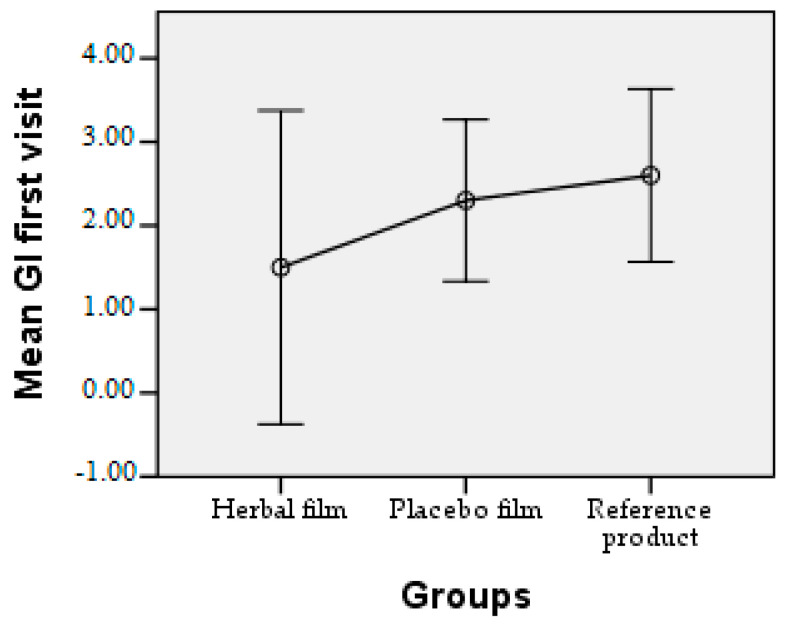
Gingival index (GI) at first visit. Patients who started treatment with the plant extract film (herbal film) had a mean GI value of 1.50, in contrast to patients who followed treatment with a reference product and in whom the mean GI was above 2.50.

**Figure 10 molecules-28-04002-f010:**
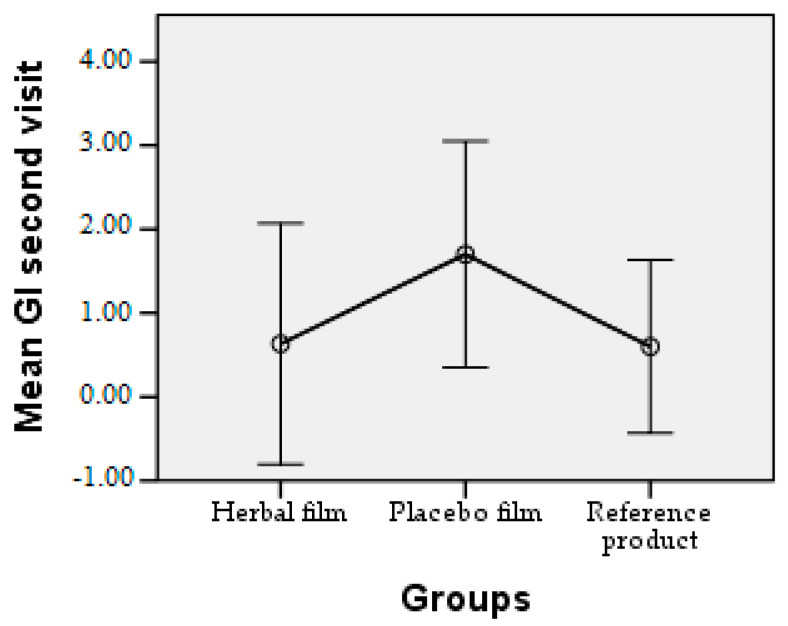
Gingival index at second visit. The difference in GI means between patients treated with the plant extract film and patients treated with the reference product is insignificant (below 0.75), in contrast to the placebo film, where the mean gingival index is higher (above 1.50).

**Figure 11 molecules-28-04002-f011:**
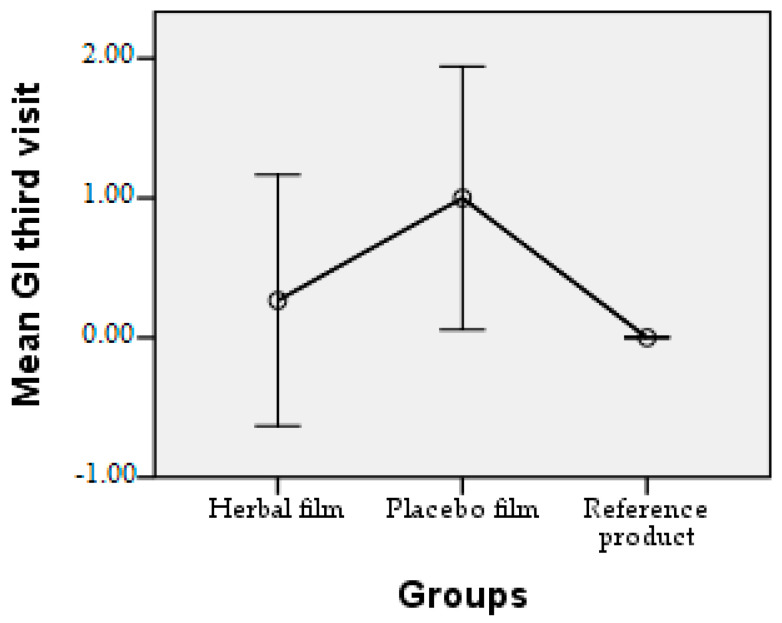
Gingival index at third visit. At the end of treatment, the mean GI decreased significantly from the time of treatment initiation, with all samples used, the mean being below 1.00.

**Figure 12 molecules-28-04002-f012:**
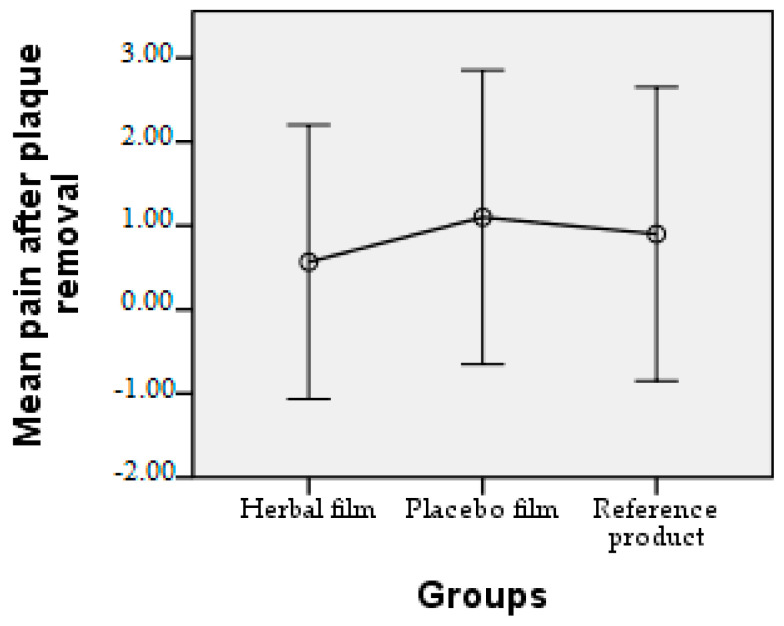
Pain after plaque removal. In case of the herbal film, the mean of the pain felt was below 0.60, unlike the placebo film with an average of 1.10. With a value close to the two films the reference product (0.90) is included.

**Figure 13 molecules-28-04002-f013:**
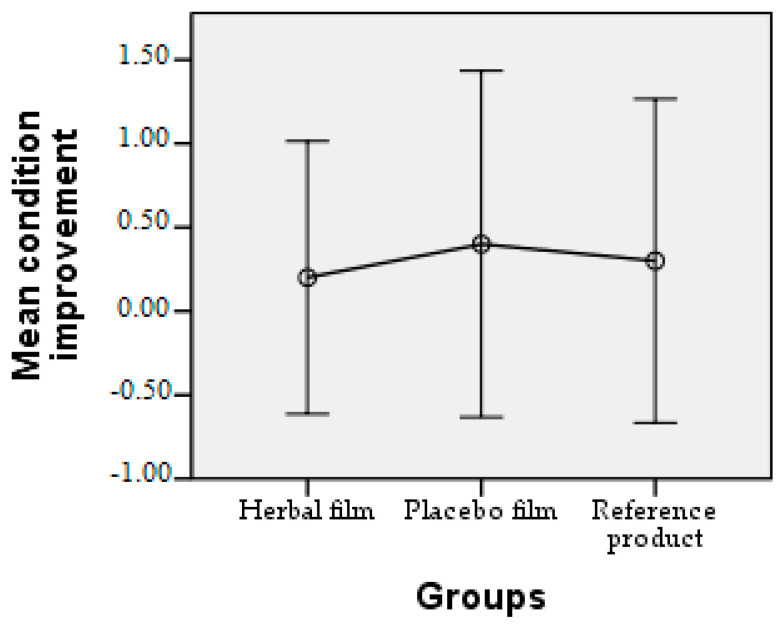
Range of improvement of the condition. For the time to improvement of inflammation, a very good average was obtained for the herbal film and a slightly higher average for the placebo film.

**Figure 14 molecules-28-04002-f014:**
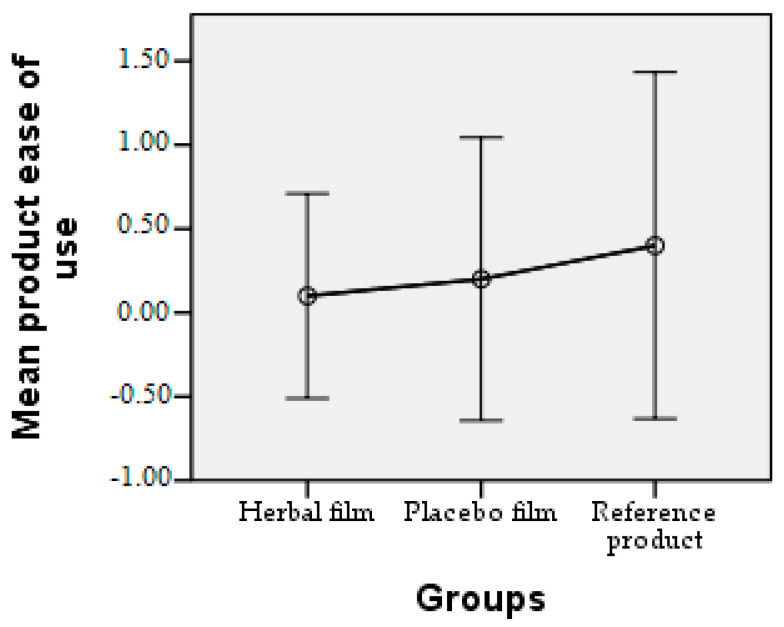
Product ease of use. It can be seen that the polymeric films were easier to use, with mean values between 0.10 and 0.20, in contrast to the reference product with a mean of 0.40.

**Figure 15 molecules-28-04002-f015:**
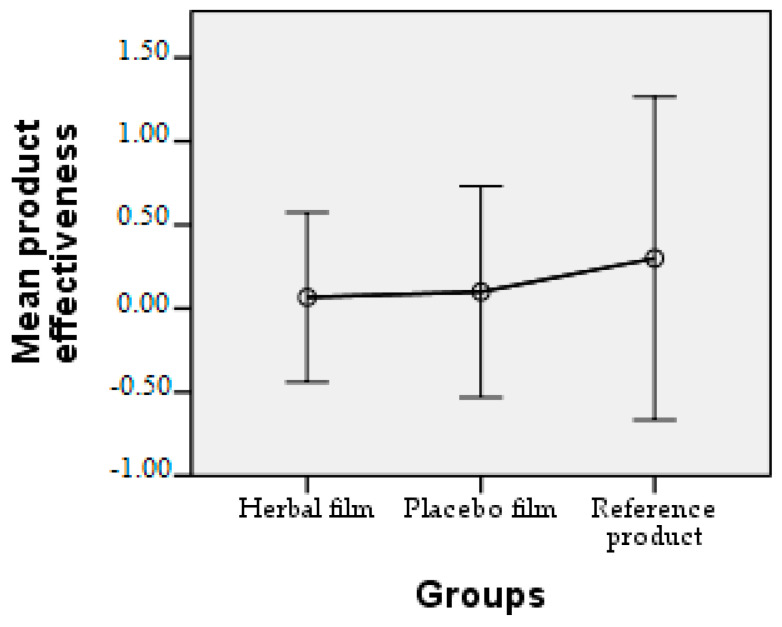
Product effectiveness. The efficacy of the polymeric films was characterised by a mean below 0.10 and the reference product achieved a mean of 0.30 for efficacy.

**Figure 16 molecules-28-04002-f016:**
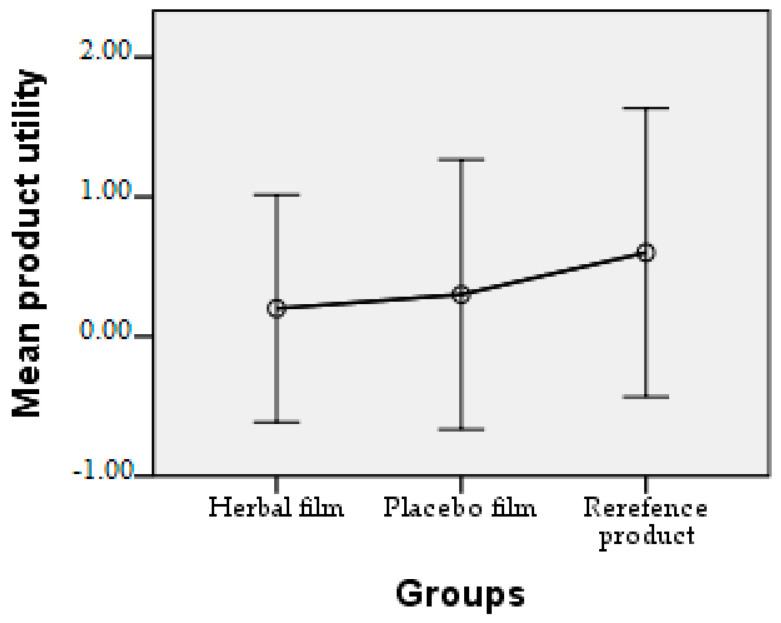
Product utility. It can be seen that the usefulness of the polymeric films is very good, with an average value below 0.30, while the reference product was rated with an average of 0.60.

**Figure 17 molecules-28-04002-f017:**
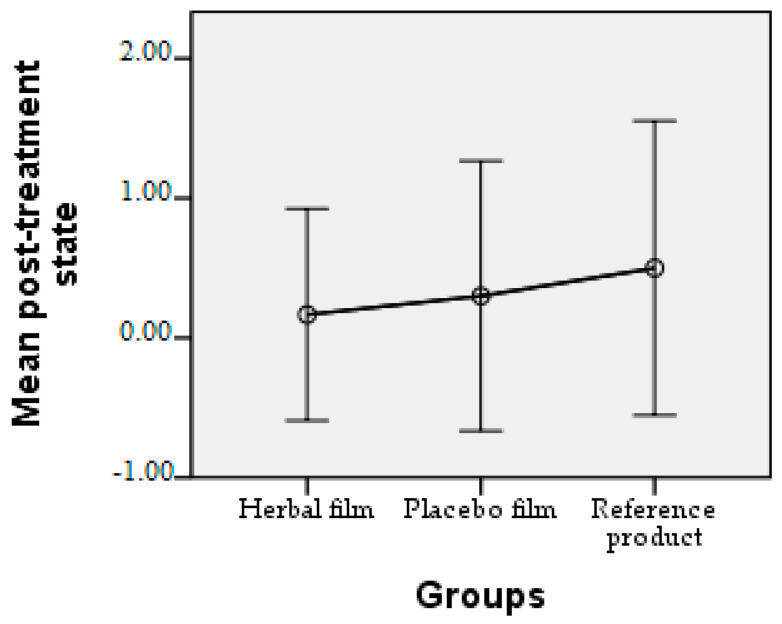
Post-treatment state. Patients rated their post-treatment state as very good, with the mean values for the three products used being below 0.5.

**Figure 18 molecules-28-04002-f018:**
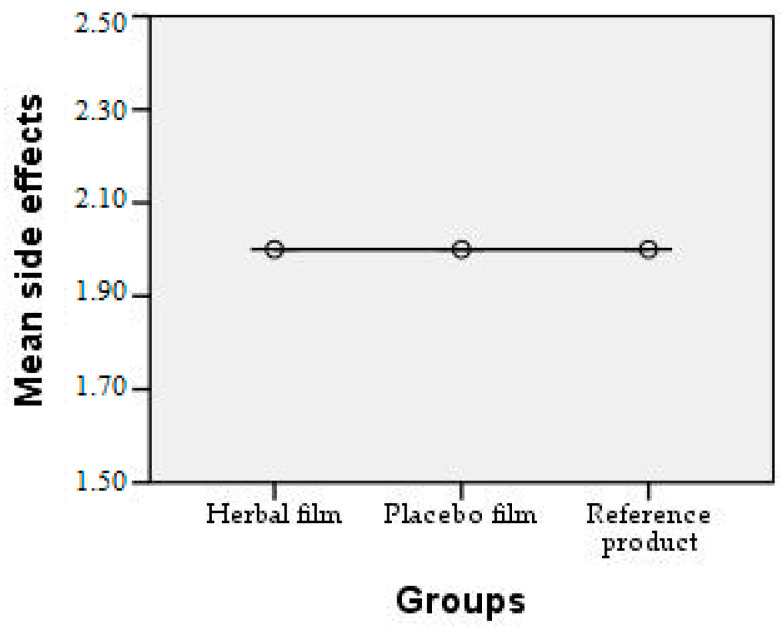
Side effects observed. At the end of the treatment no adverse reactions were observed for any of the products used.

**Table 1 molecules-28-04002-t001:** Total amount of polyphenols and total amount of flavonoids in extracts of Matricariae flos (M), Taraxaci folium (T) and mixtures.

Method	TFC mg QE/100 g DW *	TPC mg GAE/100 g DW *
Sample		
M	1.08 ± 0.054	96.28 ± 4.814
T	1.27 ± 0.063	99.50 ± 4.975
T/M 1:1	1.29 ± 0.064	98.58 ± 4.929
T/M 1:2	1.25 ± 0.062	97.72 ± 4.886
T/M 2:1	1.26 ± 0.063	98.44 ± 4.922

* Concentrations were expressed as mean ± SD, n = 3.

**Table 2 molecules-28-04002-t002:** Amount of active ingredients in extracts of Matricariae flos (M) and Taraxaci folium (T).

	M	T
**Phytosterols (µg/mL Extract) ***		
**Ergosterol**	0.187 ± 0.019	2.638 ± 0.027
**Stigmasterol**	29.260 ± 1.463	40.613 ± 2.031
**β-Sitosterol**	375.173 ± 18.758	422.233 ± 21.111
**Campesterol**	9.102 ± 0.455	2.358 ± 0.118
**Tocopherols (ng/mL extract) ***		
**α-tocopherol**	134.50 ± 0.02	91.20 ± 0.02
**γ-tocopherol**	152.60 ± 0.01	18.10 ± 0.01
**δ-tocopherol**	27.00 ± 0.01	-
**Methoxylated flavones (ng/mL) ***		
**Eupatorin**	394.97 ± 0.11	-
**Casticin**	30766.19 ± 9.52	
**Hispidulin**	1844.71 ± 2.25	108.10 ± 1.08
**Polyphenols (µg/mL) ***		
**Chlorogenic acid**	70.686 ± 8.850	54.153 ± 7.005
**Luteolin**	2.680 ± 0.033	2.818 ± 0.301
**Ferulic acid**	0.557 ± 0.048	-
**Quercitrin**	87.301 ± 6.105	-
**Quercetol**	3.642 ± 0.021	-
**Apigenin**	33.613 ± 2.057	-
**Caftaric acid**	-	16.849 ± 2.162
**p-Coumaric acid**	-	0.441 ± 0.020
**Rutozid**	-	0.857 ± 0.015
**Phenolic acids (µg/mL) ***		
**Syringic acid**	0.23 ± 0.05	0.12 ± 0.01
**Protocatechuic acid**	1.54 ± 0.11	0.40 ± 0.02
**Vanillic acid**	2.34 ± 0.13	-

* Concentrations were expressed as mean ± SD, n = 3.

**Table 3 molecules-28-04002-t003:** Antioxidant activity of extracts.

Method	DPPH Inhibition % (TE mg/mL) *	Cuprac µmol Trolox/100 µL *
Sample		
M	88.92 ± 4.446	5.20 ± 0.260
T	76.07 ± 3.803	4.20 ± 0.210
T/M 1:1	80.00 ± 4.000	4.50 ± 0.220
T/M 1:2	83.92 ± 4.196	4.70 ± 0.235
T/M 2:1	79.10 ± 3.955	4.80 ± 0.240

* Concentrations were expressed as mean ± SD, n = 3.

**Table 4 molecules-28-04002-t004:** Mucoadhesive film characteristics.

Polymeric Film	Composition	Aspect	Colour	Smell	Taste	Surface Texture
M1	PVA 5%	Uniform Homogeneous	Translucent	Odourless	Very pleasant	Smooth Non sticky
M2	PVA 8%	Uniform Homogeneous	Translucent	Odourless	Very pleasant	Smooth Non sticky
C1	PVA 5% Extract 5%	Uniform Homogeneous	Translucent Yellow	Odourless	Pleasant	Smooth Non sticky
C2	PVA 5% Extract 10%	Uniform Homogeneous	Translucent Yellow	Odourless	Pleasant	Smooth Non sticky
C3	PVA 8% Extract 5%	Uniform Homogeneous	Translucent Yellow	Odourless	Pleasant	Smooth Non sticky
C4	PVA 8% Extract 10%	Uniform Homogeneous	Translucent Yellow	Odourless	Pleasant	Smooth Non sticky

**Table 5 molecules-28-04002-t005:** Chromatic classes codes and their false rendering colour.

Chromatic Class	Chromatic Legend
**ExtrCL1**	
**ExtrCL2**	
**ExtrCL3**	
**ExtrCL4**	
**M1_M2**	
**Artefact**	

**Table 6 molecules-28-04002-t006:** Imagistic results. Proportions distribution of chromatic classes for each film sample (data are presented as means of all parts from the same film sample).

	Chromatic Class Proportions	ExtrCL1 (%)	ExtrCL2 (%)	ExtrCL3 (%)	ExtrCL4 (%)	M1_M2 (%)	Artefact (%)
Sample							
C1		0.049	0.776	8.971	89.786	0.415	0.000
C2		0.023	0.118	0.920	74.894	23.997	0.019
C3		0.013	0.972	8.812	89.920	0.265	0.012
C4		0.059	1.305	11.756	86.555	0.321	0.004
M1		0.001	0.007	0.004	0.027	99.925	0.002
M2		0.005	0.002	0.004	0.005	99.939	0.010

**Table 7 molecules-28-04002-t007:** Univariate analysis of chromatic parameters. Data are presented as mean with standard deviation.

Sample	L* (a.u.)	a* (a.u.)	b* (a.u.)	Yi (%)	Bi (%)	Observations (Pixels)
C1	86.954 c ±5.150	−12.048 a ±1.598	−5.174 b ±9.882	40.183 a ±11.965	8.544 c ±9.981	147,191
C2	91.455 b ±3.589	−10.279 d ±1.698	−22.011 b ±8.281	19.330 d ±10.548	32.744 a ±41.273	187,827
C3	87.273 c ±5.378	−11.411 b ±1.332	−8.934 b ±7.478	35.889 b ±9.364	4.838 d ±8.580	171,539
C4	86.110 c ±5.385	−10.877 c ±1.527	−9.740 b ±7.032	34.932 c ±8.942	14.553 b ±18.438	196,587
M1	92.828 a ±3.900	−7.954 e ±1.509	−41.663 a ±1.554	3.375 f ±4.291	0.000 e ±0.000	685
M2	89.314 c ±5.380	−7.498 e ±1.590	−42.536 a ±1.510	13.594 e ±15.348	0.000 e ±0.000	31

Note: Different letters, that accompanies the means values, indicate a statistically significant difference (*p* = 0.05) between the film samples. (a–e—statistical markers) Pairwise multiple comparisons were performed with post hoc Tukey’s test (*p* = 0.05), within the one-way ANOVA analysis (*p* = 0.05). The number of replicates for each film sample is presented in last column of Table 6.

**Table 8 molecules-28-04002-t008:** pH values of polymer films.

Polymeric Film	pH *	Polymeric Film	pH *
M1	7015 ± 0.005	M2	7016 ± 0.010
C1	6887 ± 0.004	C3	6693 ± 0.006
C2	6667 ± 0.004	C4	6634 ± 0.003

* Concentrations were expressed as mean ± SD, n = 3.

**Table 9 molecules-28-04002-t009:** Disintegration pace of polymeric films.

Polymeric Film	Time (s) *	Polymeric Film	Time (s) *
M1	225 ± 5	M2	245 ± 10
C1	235 ± 10	C3	255 ± 10
C2	220 ± 5	C4	260 ± 10

* Concentrations were expressed as mean ± SD, n = 3.

**Table 10 molecules-28-04002-t010:** Antioxidant activity of polymeric films.

Polymeric Film	Inhibition Percentage (%) *
C1	32.149 ± 1.71
C2	39.932 ± 1.95
C3	15.059 ± 0.75
C4	16.751 ± 0.78

* Concentrations were expressed as mean ± SD, n = 3.

**Table 11 molecules-28-04002-t011:** Active ingredient content of polymeric films.

Polymeric Film	QE (mmol/L) *	Active Ingredient Content (%) *
C1	0.422 ± 0.020	64.81 ± 3.25
C2	0.565 ± 0.033	88.89 ± 4.16
C3	0.252 ± 0.012	36.11 ± 1.82
C4	0.521 ± 0.025	81.48 ± 4.05

* Concentrations were expressed as mean ± SD, n = 3.

**Table 12 molecules-28-04002-t012:** The non-linear regression results of the film samples extract release (%).

Release (%)	Allosteric Sigmoidal			
Best-Fit Values	C1	C2	C3	C4
Vmax	94.46	84.93	91.72	88.74
h	0.9036	1.423	0.9118	1.253
Khalf	15.36	8.468	10.41	10.92
Kprime	11.80	20.92	8.467	19.98
Std. Error				
Vmax	1.710	0.5197	0.8915	1.170
h	0.05305	0.05750	0.03651	0.08798
Khalf	0.9140	0.1951	0.3306	0.4989
Kprime	1.381	2.680	0.6551	4.091
Goodness of Fit				
Degrees of Freedom	30	30	30	30
R square	0.9938	0.9963	0.9973	0.9888
Adjusted R square	0.9934	0.9961	0.9971	0.9880

**Table 13 molecules-28-04002-t013:** Percentage of PVA and plant extract in the composition of the formulated polymeric films.

	PVA %	Extract %		PVA %	Extract %
M1	5.00	0.00	M2	8.00	0.00
C1	5.00	5.00	C3	8.00	5.00
C2	5.00	10.00	C4	8.00	10.00

## Data Availability

Not applicable.

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
