# Peer review of "Formulation and Characterization of Mucoadhesive Polymeric Films Containing Extracts of Taraxaci Folium and Matricariae Flos"

_molecules, 2023, doi:10.3390/molecules28104002_

Round 1

Reviewer 1 Report

In this research paper, the authors reported a new mucoadhesive polymeric films containing extracts of Taraxaci folium and Matricariae flos for biomedical applications in gingivitis.

1.  The introduction section is very well written and explains the need of the article very nicely. Still the article lacks the actual explanation of the research novelty. It is advisable to give a novelty statement of the research.   

2.  What is the reason for the selection of a combination of extracts from chamomile flowers and dandelion leaves for gingivitis treatment? Author should compare the individual extracts from each plants.

3. In the conclusion authors should compare the antioxidant activity among the individual and combination of extracts.

Author Response

The answers can be found in the attachment.

Reviewer 2 Report

MS is interesting and presented well, but it needs some modifications before publication.

1. Novelty of the MS needs to be reflected in the introduction.

2. There are only nine references in the introduction section, more literature should be covered in the introduction section.

3. No statistical analysis to test the significant differences between results has been done for the results shown in Table 1 - Table 3.

4. The significantly different data should be marked in Figure 2.

5. Page 15, "R2 = 0.9966" it should be R2  

6. The HPLC plots for the standard and sample should be provided in the supplementary file.

Minor editing of English language required; spelling and punctuation 

Author Response

(The authors gave the same response as above.)

Reviewer 3 Report

The study is quite limited not in the data provided or in the methods used, it should deepen more the differences found in the preparations (in what differ the different extracts). The number of patients used in the trial should be considered in higher numbers to be statistically significant the actual sample is very large. Studying the effects of different concentrations and possible synergies would be desirable. How do they affect matrices? What is really the active ingredient? The advice is to evaluate different preparations (different origins) and identify a matrix that can be considered the matrix standard.

Author Response

(The authors gave the same response as above.)

Reviewer 4 Report

Dear Authors,

The Article entitled “Formulation and characterization of mucoadhesive polymeric films containing extracts of Taraxaci folium and Matricariae flos” is really an interesting one. The fallowing are few suggestion to improve the article a bit

1.      Authors should provide the novelty statement of the research in the introduction section

2.      Provide the reason for the selection of chamomile flowers and dandelion leaves over other herbal therapy

3.      Authors should provide the TEM images for better understanding of the Film texture rather than the optical microscopic image.  

4.      Authors claim that “According to other studies, mucoadhesive films with herbal extracts exhibited green coloration and smooth texture” but the figure 1 do not support the claim. Authors should provide the normal image of the formulated film for better clarity of the claim.

5.      What is the pharmacokinetics of the invitro release of the active ingredients from the film?

6.      What is the dose selection criteria for the human study?

7.      Conclusion do not support the Novelty claim. Needed to be rewritten 

A moderate check of the English grammar is needed . Specifically in the introduction and method section 

Author Response

(The authors gave the same response as above.)

Round 2

Reviewer 4 Report

Dear Authors 

Check the spellings and grammar before submission 

Spelling and grammar check is needed